# Long-term Mammography Utilization after an Initial Randomized Intervention Period by all Underserved Chilean Women in the Clinics

**DOI:** 10.3390/cancers14153734

**Published:** 2022-07-31

**Authors:** Klaus Puschel, Andrea Rioseco, Gabriela Soto, Mario Palominos, Augusto León, Mauricio Soto, Beti Thompson

**Affiliations:** 1Department of Family and Community Medicine, School of Medicine, Pontificia Universidad Católica de Chile, Santiago 8320000, Chile; kpuschel@med.puc.cl (K.P.); arioseco@uc.cl (A.R.); mgsoto1@uc.cl (G.S.); mariopalom.8@gmail.com (M.P.); msotod@gmail.com (M.S.); 2Department of Surgical Oncology, School of Medicine, Pontificia Universidad Católica de Chile, Santiago 8330024, Chile; aleon84@gmail.com; 3Cancer Prevention Program, Fred Hutchinson Cancer Research Center, 1100 Fairview Avenue N. P.O. Box 19024, Seattle, WA 98109, USA

**Keywords:** breast cancer, mammography, low socioeconomic status, RE-AIM framework, Latin America, Chile

## Abstract

**Simple Summary:**

Chile has one of the highest rates of breast cancer in Latin America. In Chile, underserved women have lower rates of mammography screening than their medium-to-high-level counterparts and higher morbidity and mortality rates of breast cancer. After a successful randomized controlled trial of low-socioeconomic-status women in a primary care clinic, we used the RE-AIM (Reach, Effectiveness, Adoption, Maintenance) framework to establish the long-term effects of that intervention. After ten years, women at the low-SES intervention clinic continued to show higher mammography rates compared to women at middle-SES clinics; further, these results continued to be above the national rates for low-socioeconomic-status peers. The RE-AIM framework indicates some of the factors that may have contributed to this successful long-term effect among marginalized women in Chile.

**Abstract:**

Chile has one of the highest rates of breast cancer in Latin America. Mammography rates among women, especially those of low socioeconomic status (SES), are thought to contribute to high breast cancer morbidity and mortality. A successful randomized controlled trial among women aged 50 to 70 in a low-SES primary care clinic in Chile led to a significant increase in mammography screening rates in a two-year intervention trial. This study assesses the sustainability of the intervention after ten years and identifies factors that might have been associated with a long-term effect using the RE-AIM (Reach, Effectiveness, Adoption, Implementation, Maintenance) framework. The mammography rates among women aged 50 to 70 in the low-SES intervention clinic were compared to two populations of women aged 50 to 70 from middle-SES clinics and to national data. Qualitative data were used to answer questions of adoption, implementation, and maintenance, while quantitative data assessed the reach and effectiveness. After ten years, low-SES women at the intervention clinic maintained significantly higher mammography screening rates vs. middle-SES women at the comparison clinics (36.2% vs. 30.1% and 19.4% *p* < 0.0001). Women of a low SES at the intervention clinic also had significantly higher screening rates compared to women of a low SES at a national level (44.2% vs. 34.2% *p* < 0.0001). RE-AIM factors contributed to understanding the long-term difference in rates. Mailed contact, outreach interventions, and the integration of health promoters as part of the Community Advisory Board were important factors associated with the effects observed. This study provides information on factors that could contribute to reducing the social gap on breast cancer screening.

## 1. Introduction

Breast cancer continues to be a leading public health problem in Latin America and represents 28% of all new cancer cases in the continent [1]. The problem is especially severe in Chile, which has one of the highest standardized breast cancer incidence rates (37.4/100,000) of all countries in Latin America [1]. More concerning is that breast cancer rates in Chile have increased by about 60% in the past three decades [2]; further, most breast cancers are detected in advanced stages [3,4,5]. Similar to breast cancer incidence, breast cancer mortality is very high in Chile (10.2/100,000) [6].

Increasingly, the breast cancer problem in Chile is one of low-socioeconomic-status (SES) women. Of cases detected by symptoms, about 60% of breast cancer cases are found in low-SES women compared to 40% found in medium-to-high-SES women [4]. There is a strong SES gap in breast cancer detection and mortality in Chile, with little systematic attention being paid to screening for early breast cancer detection. Much of this is due to the lack of a national screening program [7].

Despite the strength of the primary care network in Chile [8], there have been no systematic interventions, screening registries, or follow-up strategies that promote breast cancer screening. There are, however, national Guidelines for Breast Cancer Prevention [9]. In 2008, a National Guideline recommended a population-based screening approach. In response, by 2010, mammography screening was gradually introduced into primary care settings. As a consequence, not only was mammography screening free, full coverage of breast cancer treatment also became free in Chile [9].

In an effort to reduce the discrepancy in breast cancer incidence and mortality in low-SES women, and in conjunction with the new Guidelines, we embarked on a pilot program to increase mammography rates among women in the Catholic University (*Católica*) primary care network in the southeast area of Santiago, Chile [9]. Building on the concept that the availability of mammography is not sufficient to increase mammography utilization, we developed a comprehensive program to encourage low-SES women to seek mammography screening. The study has been described in detail elsewhere [9,10]; basically, we randomly assigned women to standard opportunistic care, a low-intensity intervention that consisted of brief advice and a personalized mailed contact, and a high intensity intervention that consisted of brief advice, a personalized mailed contact, and a telephone outreach contact if women did not adhere to mammography screening. The outcome was mammography screening as determined by electronic medical records. At the end of the intervention, the screening rates were 6% in the standard group, 51.8% in the low-intensity group, and 70.1% in the high-intensity group. 

It has now been 10 years since that pilot study, and we wished to establish whether the overall impact of the program was sustained over time. Here, we use the RE-AIM framework to discover the long-term effects of the mechanisms built into the intervention clinic during the pilot study. We compare those rates with the rates in two comparison clinics that did not have the intervention. We review what strategies endured and how the rates of mammography in this low-SES clinic now compare with those of the two other university primary care clinics that have higher SES rates, as well as how the rates compare with the national data on breast cancer screening and SES. Specifically, we look at mammography rates in all three venues in 2010, 2013, 2017, and 2020.

## 2. RE-AIM Conceptual Framework

The RE-AIM framework was developed because of the observation that the translation of science into policy and impact is both slow and inequitable [11]. Rather than focus merely on the outcomes of an intervention, RE-AIM addresses five main issues. These are Reach, Effectiveness, Adoption, Implementation, and Maintenance. 

Reach refers to the number and representation of the population participating. For example, in our pilot project, we selected a random sample of 500 women aged 50 to 70 years from a population of approximately 1700 women. Of the random sample, we were able to collect clinic outcome data on all 500 women, indicating the good representation of the population of that type in the clinic. However, RE-AIM specifies that additional information be known about the participants such as their characteristics, qualitative reasons why they participated, and so forth. For this, we used an end-of-study survey that was completed by 93.4% of the standard group, 92.2% of the low-intensity group, and 91.6% of the high-intensity group. Thus, we have high confidence that our understanding of the women’s characteristics was good. We have similar data on the women in the comparison clinics and those in the national dataset. We are now interested in the reach of the entire primary care clinic, as well as the comparison clinics; thus, we will examine the mammography screening rate among women aged 50 to 70 years at various intervals after the intervention ended.

Effectiveness indicates the impact of an intervention, and, as previously reported, our effectiveness was high, with a 51% to 70% adherence to mammography compared to 6% for the comparison group. Effectiveness also refers to larger-scale effectiveness, and we are now interested in how the effectiveness rate in our intervention clinic compares to similar clinics within *Católica*, the same primary care network. 

Adoption refers to intervention agents who deliver the program. Typically, this is accomplished through process evaluation, if at all, and current adoption criteria include representation at the clinic level. Again, we had great adoption by the physicians and staff members in our initial work. What is now important is to ascertain the degree to which our physicians and staff continued the intervention activities. This will be another focus of this work.

Implementation is the extent to which the adoption achieved fidelity; that is, whether the intervention deliverers delivered the intervention as it was planned or piloted. Again, we were confident this occurred in the pilot study, but it is important to note not only the extent to which fidelity occurred but also the changes that were made to implementation after the intervention period. An examination of what remains of the intervention activities will help us answer this question.

Finally, maintenance refers to the extent to which the behavioral change—mammography utilization—continues over time and whether the clinics institutionalized policies or practices to continue the intervention or aspects of it. Qualitative data will help us understand this question; however, the major impact will be determined by the difference in the mammography utilization in this low-SES clinic compared to clinics in the same network, as well as national rates of mammography utilization.

## 3. Materials and Methods

The major hypothesis under examination in this study is as follows: the women aged 50 to 70 years in the 2010 low-SES intervention clinic will have higher rates of mammography utilization over time compared to the women in the two medium-to-high-SES comparison clinics in the same network and compared to low-SES and medium-SES women nationally. In addition, we will examine the RE-AIM dimensions that can facilitate or impede long-term achievement in equitable impact.

### 3.1. Description of the Clinics

The Intervention Clinic is located in El Castillo, La Pintana, Chile, an area with a relatively young population of very low socioeconomic status. El Castillo is a very active and consolidated community with many local organizations in place. The Intervention Clinic developed a community advisory board (CAB) in El Castillo simultaneously with the inauguration of the clinic in 2006. The local community assisted in many clinic-focused health prevention and promotion initiatives, including the above-referenced randomized breast cancer screening program, which occurred between 2008 and 2010. 

The Comparison Clinics are located in East Puente Alto, Chile, an area of higher socioeconomic status than El Castillo, with a slightly older population. The local communities in Puente Alto are more heterogeneous, less integrated, and have higher levels of education than those in El Castillo. As part of their health care model, all primary care clinics are required to have a community advisory board that participates in the community assessment of local health needs, helps to develop annual plans, and supports the interaction between clinics and local communities [12]. Thus, both comparison clinics have also had CABs since the inauguration of the clinics. 

All three primary care clinics included in the randomized study are university clinics, sponsored by *Católica,* that provide free healthcare to a population of about 20,000 people each—all of low-to-medium socioeconomic status. They are located in underserved areas in the southeast area of Santiago, Chile. The clinics are administered by *Católica* but belong to the public primary care network and are funded by the government based on the same per capita system that funds the entire primary care system in the country. Individuals need to register at the clinic to receive care. In that manner, the clinics serve a limited and stable population, and they are integrated to the local communities. The services provided by the clinics are defined by law. They include preventive services (e.g., immunizations, well-childcare, and regular check-ups), mental and social services, and clinical services (acute medical care, chronic health care, dental care, and physical therapy). All primary care clinics need to present an annual plan based on national primary care guidelines [13]. 

### 3.2. Data Gathered from the Clinics

The rates of mammography screening were gathered from electronic medical records in each of the three clinics in each of the years referenced. The qualitative data gathered from the clinics during the randomized study were collected via surveys and key informant interviews in 2008 and at the end of the study in 2010 [9]. Post-intervention qualitative data (2013, 2017, and 2020) were collected via key informant interviews within each organization at the appropriate time intervals. Key informants included nurses, midwives, clinic directors, clinic administrators, and CAB members.

### 3.3. National Data Source

The national data coverage from 2009–2010 and 2016–2017 was obtained from the Chilean National Health Surveys [14]. The surveys followed standard international WHO criteria [15] and were based on a representative national sample of people aged 14 years and older selected randomly in a multi-stage process conducted in all regions of the country. In-person interviews were conducted by health professionals (trained interviewers and nurses) with participants in their households to assess 72 health problems and risk factors. The assessment included self-reported information, physical examinations, and blood and urine samples. In the 2017 version of the National Survey, 6233 individuals participated. The collection of information and the analysis took about six months. Mammogram screening was assessed based on self-reporting by asking patients if they have had a mammogram during the last three years. Although the 2017 survey is the most recent to date, a new version of the National Health Survey is planned within the next two years. 

## 4. Results

The results are presented as being consistent with the RE-AIM framework.

### 4.1. Reach

A total of 1700 women aged 50 to 70 were included in the intervention clinic (see Table 1), compared to 4832 in the comparison clinics. Although only a subset of women in the intervention clinic were selected (*n* = 500), the total reach included all eligible-aged women in the intervention and comparison clinics. From Table 1, it can be seen that the characteristics of the subsample women were comparable to the characteristics of all the women in the intervention clinic. Using education as a proxy for socioeconomic status, we see that women in the intervention clinic had a much lower educational attainment than those in the comparison clinics. Further, when examining the national data, the intervention women had much lower educational attainment. In terms of poverty, the intervention women had significantly higher poverty levels than the comparison or national women. They were also more likely to have higher overall mortality and mortality from Covid-19.

After the intervention concluded in 2010, the women in the intervention arm who participated in the study reported a significant increase regarding where and how to get a free mammogram; this was not seen in the comparison arm. In addition, the women who participated in the high intensity arm reported feeling less embarrassed to get a mammogram than the women in the comparison group. 

### 4.2. Effectiveness

The trial was effective with 51.8% of women in the low-intensity arm and 70.1% of women in the high-intensity arm having a mammogram. For the effectiveness in the intervention clinic overall, Figure 1 indicates the rate of mammography screening in Years 2010, 2013, 2017, and 2020 for the overall low-SES intervention clinic and the two comparison clinics that had higher SES rates. As can be seen, the intervention clinic continued to have higher rates of mammography after the trial ended in 2010, with roughly similar patterns in 2013 and 2017 and with a slight decline in 2020, when Covid-19 was prevalent.

### 4.3. Adoption

Table 2 indicates what continued of the intervention after it ended in the intervention clinic and what was taken on by the comparison clinics. The adoption of the intervention occurred at two levels once the RCT ended. At the organizational level, the intervention clinic moved from opportunistic screening and began to order mammograms systematically for women. This was accomplished by having midwives connect the mammography program to the Pap-testing program. This systematic approach integrated other health care professionals—midwives—to patients, relieving some of the burden of busy physicians. This core component was adopted early in the RCT and has now also been adopted in the comparison clinics.

Another organizational change was the adoption of the strategy to mail an order for a mammogram and thereby facilitate screening, as was developed in the RCT. In this strategy, women received, by postal or electronic mail, a mammogram order, so they could go directly to the mammography unit without the need for a clinical visit. This strategy was another core component of the RCT intervention but was also adopted early on by the entire clinic and was also implemented afterward in the two comparison clinics.

At the individual level, local social media played a large role in adoption by age-eligible women. “Whatsapp” and “Instagram” have emerged in communities and are associated with primary care clinics. These networks were used to inform and communicate with women regarding the importance of mammograms as well as practical ways to obtain a mammogram. The use of social media was initiated at the intervention clinic and adopted early on. Again, this has subsequently been implemented in the comparison communities.

### 4.4. Implementation

As can be seen from the adoption section, many of the successful strategies that were adopted as part of the randomized intervention were later adopted by and implemented in the comparison clinics. The systematic screening implemented by midwives was likewise done at comparison clinics in conjunction with the Pap-testing program; thus, there was the fidelity of the intervention. The same was true for the letters. There may have been some differences in the social media posts and information, but we were unable to review all the posts.

Implementation in the intervention clinic was greatly influenced by community-based changes. All clinics that are part of the *Católica* system are required to have a Community Advisory Board (CAB) to work with clinic providers and staff. Breast cancer screening was not a relevant topic for the CAB, initially. One of the health volunteers who was an early participant in the project also was a CAB member and contributed greatly to make mammography screening a relevant topic for the local CAB and the community. She also helped to facilitate access for mammogram screening at the primary care clinic. She was key in changing community- and clinical-level information, which negated the generally accepted cultural idea that a mammogram is only needed if there are breast symptoms. She was a tireless worker for implementing the project. The idea of having a “champion” of mammogram screening at the CAB was adopted very early on at the intervention clinic but has been only partially implemented at the comparison clinics

### 4.5. Maintenance

Figure 1 shows the maintenance of mammography utilization in all three clinics. The mammography rates at the low-SES intervention clinic decreased over time but remained significantly higher than those observed at the middle-SES comparison clinics across the years. During the Covid-19 pandemic period (2020), all of the clinics experienced a significant decrease in screening rates. An increase in mammography rates at the low-SES intervention clinic was observed between 2013 and 2017 (pre-pandemic period), which was not sustained during the pandemic period. From Table 2, we see that the comparison clinics adopted and implemented some of the successful intervention strategies in 2010, and this also increased their utilization.

By examining the mammography rates in the low-SES intervention clinic and the national data for the low-SES and medium-SES women, we see that, in 2010, the intervention clinic had substantially higher rates than the national rate for both low- and medium-SES women. By 2017, the mammography rate for the low-SES women in the intervention clinic remained higher than the national rate for the low-SES women; however, the national mammography rate for medium-SES women increased to approximately 55%, a rate that was 7% higher than that for the intervention clinic women (see Figure 2).

## 5. Discussion

In this study, we examined the long-term effects of an intervention targeted toward low-SES women aged 50 to 70 in a primary care clinic in Chile. After a successful RCT indicated that low-SES women were significantly more likely to have a mammogram than their medium-SES counterparts in comparison clinics, we used the RE-AIM framework to determine if such an intervention was adopted or translated into meaningful outcomes [16]. By comparing the intervention clinic mammography rates to the comparison clinic mammography rates, we noted a positive effect over time, with rates remaining higher in the low-SES clinic compared to those in the medium-SES comparison clinics. That differential remained through 2020. By comparing the intervention clinic with the national data, we saw that the intervention clinic continually had higher mammography rates than the low-SES national data, although the medium-SES women exceeded the low-SES women’s rates by 2017. Nevertheless, the intervention rates for the low-SES women remained higher for the intervention clinic for ten years post-intervention.

This study provides evidence of the importance of long-term effects of a primary care intervention on breast cancer screening. It contributes to filling the gap of information on the long-term effects of cancer screening programs and their impact on reducing health disparities at a clinic- and a population-based level. A systematic review conducted by Shah and colleagues [17] on community-based programs promoting breast cancer screening to reduce health disparities included 34 studies and found only two programs with a long-term follow-up defined as more than one breast cancer screening cycle. The authors emphasized the need for long-term population-based evaluations of such cancer screening programs. 

Our study shows that, after the intervention trial concluded in 2010, there was a significant decrease in mammography screening rates at the low-SES intervention clinic. This is an expected effect given that the intensity (dose) of an intervention tends to decrease when an innovation is translated and diluted into regular practice. However, the mammography screening rates at the low-SES intervention clinic remained higher compared to the middle-SES intervention clinics after ten years. Further, they were 11% higher (34% vs. 45%) compared to an equivalent population at a national level.

The mammography screening rate was 11% higher (34% vs. 45%) in the population of low-SES women at the intervention clinic compared to an equivalent population at a national level. It also was higher compared to the reference clinics of medium-SES after ten years of evaluation. The rate of mammogram screening observed in 2017 in our intervention group (45%) is higher than those reported in US women with no health insurance (39%), according to the national data provided by the American Cancer Society [18], and close to the average rates (49%) reported in East European countries [19]. The screening rates experienced a decrease in the last period, in 2020, evaluated in all clinics, probably as an effect of the pandemic. This effect has also been reported in a study conducted by the American Cancer Society [20], where a decline in the mammogram screening rate to 41% in was observed among Hispanics of low SES [18]. 

In Latin America, most countries do not have national breast cancer screening programs, or they are only partially implemented [21]. There is a high variability in breast cancer screening rates, and in many countries, less than 50% of the population has access to mammogram screening [22]. In the last National Chilean Health Survey (2017), the coverage of mammography for women between 50 and 70 years was 56%. Therefore, an important increase was observed at the national level after the introduction of the national guideline, especially in populations of medium and high SES. These are better educated groups that obtain healthcare in primary care clinics located in wealthier municipalities with fewer care demands and better access to mammogram clinics. As has been reported in other studies, national health policies tend to disproportionally affect populations of different socioeconomic groups. 

The Chilean national survey showed that there were still very significant differences by SES. Women of low SES reported mammogram screening rates of only 37%. Low educational levels have been identified as a main barrier for reducing the prevention gap in breast cancer prevention among Latin American women [23]. Our study indicates that a low-level intervention that engages community members and local health teams can reduce the prevention gap and maintain long-term changes in a population with a very low education level. 

The RE-AIM model used to assess our intervention over time is one of the most widely applied frameworks to plan and evaluate the implementation of public health interventions [24]. In a systematic review conducted by D’Lima and colleagues [24], the authors found 157 articles that applied the RE-AIM framework to assess the process and impact of public health interventions. However, only 3 of the 157 interventions were from Latin America, and none of them were related to cancer prevention. In a subset analysis of that study, the authors found that only 60% of the articles assessed the five dimensions of the framework, and less than 20% specifically identified the magnitude of the population at risk by including denominators in the estimated rates. In our study, we were able to explore each component of the RE-AIM framework and clearly identify the target population. The analysis showed that, besides the effectiveness of the intervention, the level of adoption and implementation was very high, and the intervention was included in the regular health promotion activities implemented at the clinic. The intensity of the intervention will probably need a booster to improve the screening rates in the next few years, especially after the pandemic. The relevance given recently to breast cancer prevention in national health policies provides a better scenario to reach higher screening rates and have an impact on morbidity and mortality rates in the near future [13].

### Limitations

This study is not without limitations. Despite our best intentions, not all of the components of the intervention remained in place. Further, although the comparison clinics implemented some of the evidence-based interventions, they did not include all the components. Other clinics or communities in the country might have implemented other strategies that were not captured and analyzed in this study. This might have been especially relevant for communities of higher SES that have improved their information and access to health care significantly during the last ten years and have therefore reduced their barriers to obtaining mammograms compared to communities of lower SES. This is a limitation of long-term studies that are susceptible to ecological bias. An equivalent comparison and a clear description of potential confounder variables such as age range, socioeconomic status, and educational level are presented in the study to reduce the risk of ecological bias. Another limitation is that the data for the national study were repeated only twice—once in 2010 and once in 2017. Although the rate of mammography increased by about 10 percentage points in the low-SES population, it is not clear whether that was connected to a specific event or a gradual increase in mammography rates. Despite this, it is still clear that the national rates did not reach the level of the intervention clinic rates. In terms of the RE-AIM framework, we were not able to analyze all aspects of the framework; for example, we did not have information on women’s satisfaction with the adoption of the intervention. Nevertheless, we were able to comment on the five major areas of the framework.

## 6. Conclusions

The long-term effects of an RTC in a primary care clinic in a low-SES area of Santiago, Chile was sustained ten years after the implementation of the intervention. Further, the effects were seen in the entire clinic, not only for the random sample of women who participated in the RTC. Compared to national Chilean data, the effect was sufficient and sustained enough to supersede the rates of mammography seen in low-SES women throughout the country. The use of the RE-AIM framework shed much light on the processes of the sustainability of the intervention effects. Describing all five factors of the framework was a unique feature of this work and is consistent with the goals of the framework. Future studies should examine the effects of booster sessions and implementation settings in other clinics.

## Figures and Tables

**Figure 1 cancers-14-03734-f001:**
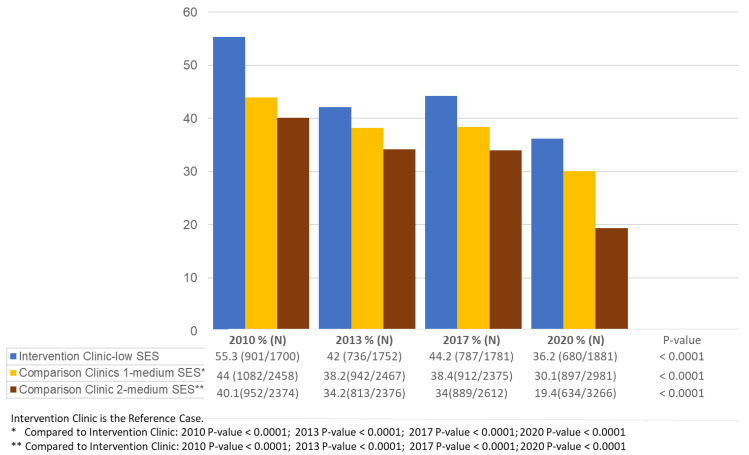
Mammography screening rates in the intervention and comparison clinics by year.

**Figure 2 cancers-14-03734-f002:**
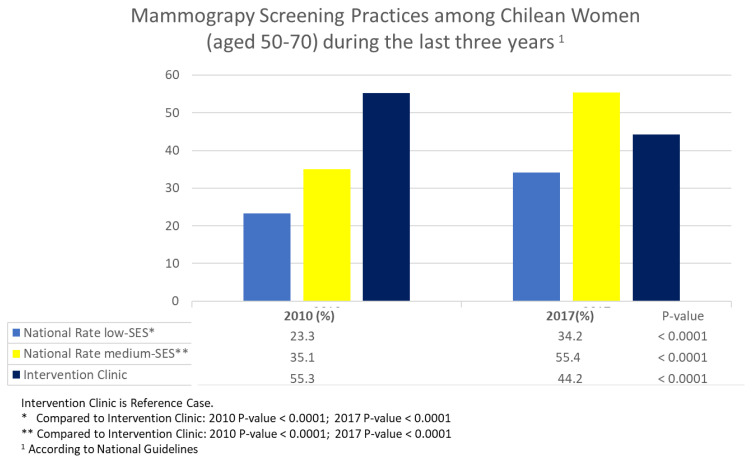
National and Intervention Clinic Data for Mammography Utilization in 2010 and 2017.

**Table 1 cancers-14-03734-t001:** Characteristics of the population in the intervention and comparison clinics.

Characteristic	Intervention Clinic	Comparison Clinics (Combined)	National Data	*p* Value ^a^
Eligible population (*n*) (Women 50–70 years)				
2010	1700	4832		<0.0001
2013	1752	4841	1627	<0.0001
2017	1781	4987		<0.0001
2020	1881	6247	2170	<0.0001
Education (%)				<0.0001
≤8 years	48.2	32.3	29.8	
9–12 years	47.5 ^b^	26.2	44.6 ^b^	
>12 years	4.3	38.1	25.6	
Percent under poverty level by income	15.3	7.8	10.8	<0.0001
Mortality rate by Covid-19per 100,000 (2020–2022)	390	296	178.9	<0.0001
Potential years of life lost per 100,000	79.5	61.0	66.1	<0.0001

^a^ *p*-values obtained by comparing the intervention clinic with the comparison clinics and national data in terms of each of the characteristics assessed. The intervention clinic was used as the reference case in all comparisons. ^b^ *p*-value = 0.036 when comparing the intervention clinic vs. the national data at 9–12 education years

**Table 2 cancers-14-03734-t002:** Activities during adoption and implementation.

Activity	Adoption/Implementation During Intervention		Adoption/Implementation Post-Intervention (2013)	
	Intervention	Comparison	Intervention	Comparison
Brief advice	Yes	Yes	Yes	Yes
Mailed contact	Yes	No	Yes	No
MD letter	Yes	No	No	No
Informational brochure	Yes	No	No	No
Mammogram order	Yes	No	Yes	No
Outreach contact	Yes	No	Yes	Yes
Telephone contact for non-adherent women	Yes	No	Yes	Yes
In-home contact, if necessary	Yes	No	No	No
Health promoter in CAB Local social media	Yes	No	Yes	Partially

## Data Availability

Not applicable.

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
