# Peer review of "Long-term Mammography Utilization after an Initial Randomized Intervention Period by all Underserved Chilean Women in the Clinics"

_cancers, 2022, doi:10.3390/cancers14153734_

Round 1
Reviewer 1 Report
Although the subject of this study is relevant, it is not new, there are others researches with the same subject and all of them has already identified this kind of situation into the low socioeconomic group, in this way the study is just a confirmation of a reality already known, however, it is always important to alert health authorities in order to maintain awareness among the population.I am forwarding correction/suggestions for improving the presentation to the author, as well as complementing the study with further investigations
Line 92 - “Reach refers to the number and representation of the population participating we selected a random sample of 500 women aged 50 to 70 years of age from a population of approximately 1400 women”.
However, in the line 187 “A total of 1700 women aged 50 to 70 was included in the intervention clinic”. It seems like a conflict between the number of women.
Line 218 - The acronym RCT, what it means?In table 1 is not very clear the values in column “National data”
In table 1 the results presented show a higher rate of mammography for the clinical intervention group.I would like your comments regarding the mortality rate for this group.
It would be important to include statistical analysis for better consistency in the results.
Reviewer 2 Report
This paper presents a long term (10 year) follow up of an experiment in stimulating breast cancer screening in a low SES area, by comparing an intervention clinic (and catchment area population) with 2 other clinics using usual care.
Methodologically this bears quite some issues that are not dealt with appropriately. The authors chose to present a long term follow up without explaining all intermediate developments and interventions that have undoubtedly occurred around these clinics (investments, change of leadership, improvement projects, national regulations, financial crises etc,etc.). Neither do they properly describe the size, staffing, diagnostic investment polocy etc.etc. per clinic.
Further the effectiveness data of the first trial are not presented in the same way, with exact scores and percentages on the follow up figures, nor is it clear whether variation per year is analysed. It is already very difficult to compare the raw data, let alone having a thorough statistical advice on how to establish significance (or the absence of..)
A rather raw presentation of measures/activities in table 2 does not give a proper impression of the xact intervention, nor whether these were all continued exactly in the same way over the years. From fig 1 I cannot decipher the exact numbers involved, nor whether they were corrected for target population size.
The discussion states quite straightforward that the original intervention has long lasting effects, but the reader cannot deduct that from the data provided.
The fact the a champion seemed decisive in the pattern found is not surprising, but was this part of the original intervention and what does that mean???
Moreover, the decreasing trend in all clinics (fig 1) versus the changing trend in fig 2 is not expleined well nor in text, nor statistically.
In the abstract the measures/activities of the intervention that were estimated to be effective are not mentioned.
In all a rather complete revision with a critical statistical input is needed to establish whether the paper is worth publishing; in it’s present form I would advise to reject.
Reviewer 3 Report
This manuscript describes the examination of mammography utilization in Chile using the RE-AIM framework, focusing on women 10 years after intervention. The title is difficult to understand because of two acronyms, and the meaning of the phrase “10 years after intervention” is not clear. The decreasing trend from 2010 to 2020 in Figure 1 is not well interpreted. The significance of this study is not clearly described.
Major
· Title: the use of acronyms such as SES and RE-AIM does not present a clear target for this study. The phrase “10 years after intervention” is misleading. The readers may think that this study focuses on women who had received the treatment 10 years ago.
· RE-AIM: The RE-AIM is used in the title, simple summary, abstract, etc. without any explanation.
· RE-AIM in other studies: the manuscript does not cite a sufficient number of references and the connection of this study to others, particularly other RE-AIM studies, is not fully described.
· The decreasing trend in Figure 1: this decreasing trend is stated in lines 267-268 (… rates decreased a bit in the intervention clinic, ..). This decrease is not “a bit” but significant. The authors should explain the cause of this decrease, which is not consistent with the conclusion of this study.
· Past three years in Figure 2: The phrase “in the past three years” in the title of Figure 2 needs to be explained since the bars are labeled 2010 and 2017 without referring to 3 years.
Minor
· p values in table 1: please indicate the comparisons for deriving the p values in the last column.
· abstract: the current abstract is written without any quantitative descriptions. Most sentences are vague, in particular, the last sentence.
· Chilean women: the brief introduction of Chilean women using scientific/genetic terms might be a plus.
Round 2
Reviewer 1 Report
Thank you for the answers to my questions, but I identified an inconsistency in Figure 1, as shown below:Figure 1. The percentage (901/1700) does not correspond with the percentage shown, it needs to be revised in Figure 2 as well.
Reviewer 2 Report
UNfortunately, the authors have almost not dealt with my earlier suggestions, in terms of methodological en presentation issues.
Reviewer 3 Report
The authors responded satisfactorily to the comments. Thank you.
Round 3
Reviewer 2 Report
Unfortunately the authors persist in explaining that my comments were not applicable, not relevant or that I misunderstood the nature of the circumstances described in the paper. I feel that my earlier review comments are not properly answered and therefore commend to reject the paper.